# Posterior probabilities of membership of repertoires in acoustic clades

**Hal Whitehead**\*, **Taylor A. Hersh**¤

Department of Biology, Dalhousie University, Halifax, Nova Scotia, Canada

¤ Current address: Comparative Bioacoustics Group, Max Planck Institute for Psycholinguistics, Nijmegen, Netherlands
\* hwhitehe@dal.ca

**Data Availability Statement:** The IDcall and IDcallPP code is under active development by the authors and can be accessed, along with the sperm whale datasets, through the Open Science Framework (https://osf.io/5fter/). The wren (Halfwerk et al., 2016) and cricket (Moran et al.,

## Abstract

Recordings of calls may be used to assess population structure for acoustic species. This can be particularly effective if there are identity calls, produced nearly exclusively by just one population segment. The identity call method, IDcall, classifies calls into types using contaminated mixture models, and then clusters repertoires of calls into identity clades (potential population segments) using identity calls that are characteristic of the repertoires in each identity clade. We show how to calculate the Bayesian posterior probabilities that each repertoire is a member of each identity clade, and display this information as a stacked bar graph. This methodology (IDcallPP) is introduced using the output of IDcall but could easily be adapted to estimate posterior probabilities of clade membership when acoustic clades are delineated using other methods. This output is similar to that of the STRUCTURE software which uses molecular genetic data to assess population structure and has become a standard in conservation genetics. The technique introduced here should be a valuable asset to those who use acoustic data to address evolution, ecology, or conservation, and creates a methodological and conceptual bridge between geneticists and acousticians who aim to assess population structure.

## Introduction

Many animals communicate or sense their environment using sound [1]. It is often logistically easier to record acoustic signals than to collect genetic, morphological, or other phenotypic data. Thus, the characteristics of animal calls have been used to examine a range of issues in biology, including evolution [e.g. 2], population structure [e.g. 3, 4] and conservation [e.g. 5]. Call attributes can be genetically or culturally inherited [e.g. 6, 7]. In either case, if there is drift or selection, variation in these attributes may signal population structure. This will especially be the case if the calls themselves structure populations, for instance if song attributes proscribe mate choice [e.g. 8]. Additionally, if the animals themselves use call attributes to identify segments of a population ("us versus them"), and this population structure circumscribes social interactions, and so social learning opportunities, the acoustically-distinguished population segments will tend to have distinct cultural behaviour in various contexts, including non-acoustic behaviour, such as foraging techniques [9].

2020) data can be accessed through Dryad at https://doi.org/10.5061/dryad.q5p7g and https://doi.org/10.5061/dryad.wpzgmsbhr, respectively.

**Funding:** The authors received no specific funding for this work.

**Competing interests:** The authors have declared that no competing interests exist.

Thus, there is increasing interest in using acoustic data to examine population structure. This is, however, dwarfed by molecular genetic methodologies. The majority of population structures inferred for animal species are based on genetic data, which are processed using a range of analytical methods [10]. Of these, the STRUCTURE package is particularly popular and influential [11]. STRUCTURE uses a Bayesian approach to calculate, from genetic data, posterior probabilities that individuals belong to each of $K$ source populations, or, in the admixture option, to have a proportional assignment to each of the populations [12, 13]. The results are displayed as stacked bar plots of posterior probabilities that each individual is a member of each population segment, or the estimated mixture proportions of source populations for a given individual. STRUCTURE thus gives direct estimates of the number of population segments, their distributions (in space, time, or along other axes), and confidence in allocations of individuals to the different population segments.

An analogous method of analyzing and displaying acoustic data has the potential to be similarly useful for calling animals [14]. The IDcall routine (summarized in Fig 1) uses multivariate information on calls that are grouped into repertoires. It classifies the calls into types using contaminated mixture models. Each call has a probability of being a member of each type. The repertoires of calls are then clustered into identity clades by identity call types: identity clades are marked by one or more identity calls that are made frequently by the repertoires in the identity clade and rarely by those outside it. The IDcall framework is then somewhat analogous to the initial steps of STRUCTURE: the repertoires (from individuals or groups of individuals) are classified into population segments (identity clades), with the number of population segments being determined by the routine. However, only some call types are identity calls, and some repertoires may not be assigned to an identity clade.

Here we show how the output of IDcall can be used to calculate the posterior probabilities that each repertoire is a member of each identity clade, and then display these posterior probabilities as a stacked bar graph using a routine that we call IDcallPP. A similar approach could be used with other methods of clustering acoustic repertoires to ascertain confidence in the assignment of acoustic repertoires to clusters. These outputs, especially the stacked bar graphs, parallel those of STRUCTURE.

## Methodology

### Theory

In IDcall (see Fig 1), the contaminated mixture model algorithm estimates the probability that each call, $i$, belongs to each call type, $j$, as $u(i,j)$ (where $u(i,j) = 0$ if call $i$ is characterized by a different set of variables than the calls in $j$). The usage, $U$, of each call type, $j$, for each repertoire, $r$, is calculated by summing the probability of call type membership for all calls $\{i\}$ in the repertoire and dividing by the total number of calls in the repertoire, $n(r)$:

$$U(r,j) = \frac{\sum_{i \in r} u(i,j)}{n(r)} \tag{1}$$

The following procedures in IDcallPP are summarized in Fig 2.

Once repertoires are assigned to an identity clade, $c$, we can estimate the probability distribution of call types in the identity clade as, where $r$ represents the repertoire of interest.

$$P(j,c) = \frac{\sum_{r \in c} U(r,j)}{\sum_{r \in c} \sum_k U(r,k)} \tag{2}$$

This is somewhat circular, as if repertoire $R$ was assigned membership of identity clade $c$, the call type distribution within $R$ is used to estimate the distribution of call types of $c$, which will

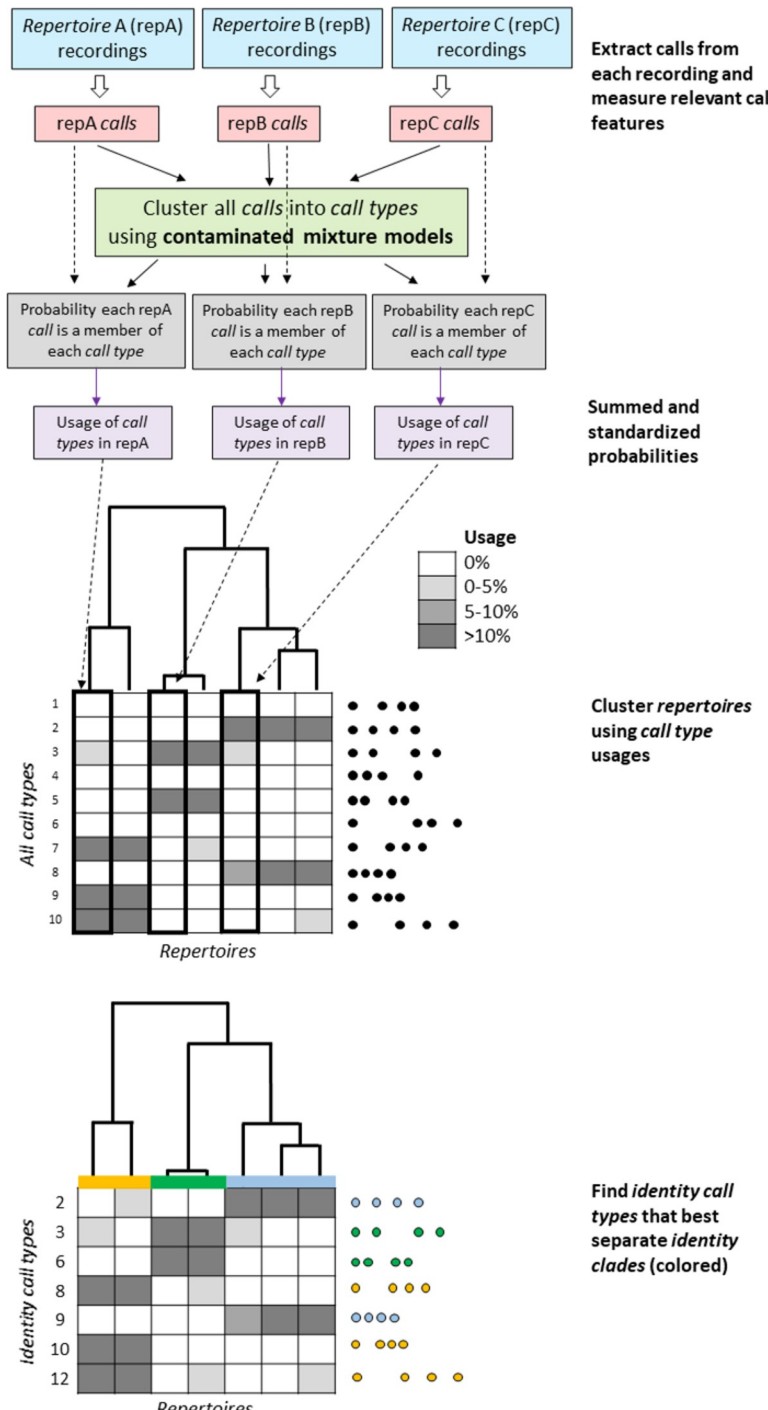

**Fig 1. The major elements of IDcall framework.** The recordings of sounds are delineated into identity clades based on the identity calls that characterize them.

then be used to calculate the likelihood that repertoire $R$ is from identity clade $c$. In other words, the calls heard in a repertoire are used to delineate identity clades, the very information that is used to calculate the posterior probability that the repertoire is a member of an identity clade. To remove this circularity, we omit repertoire $R$ from the calculation of the call type

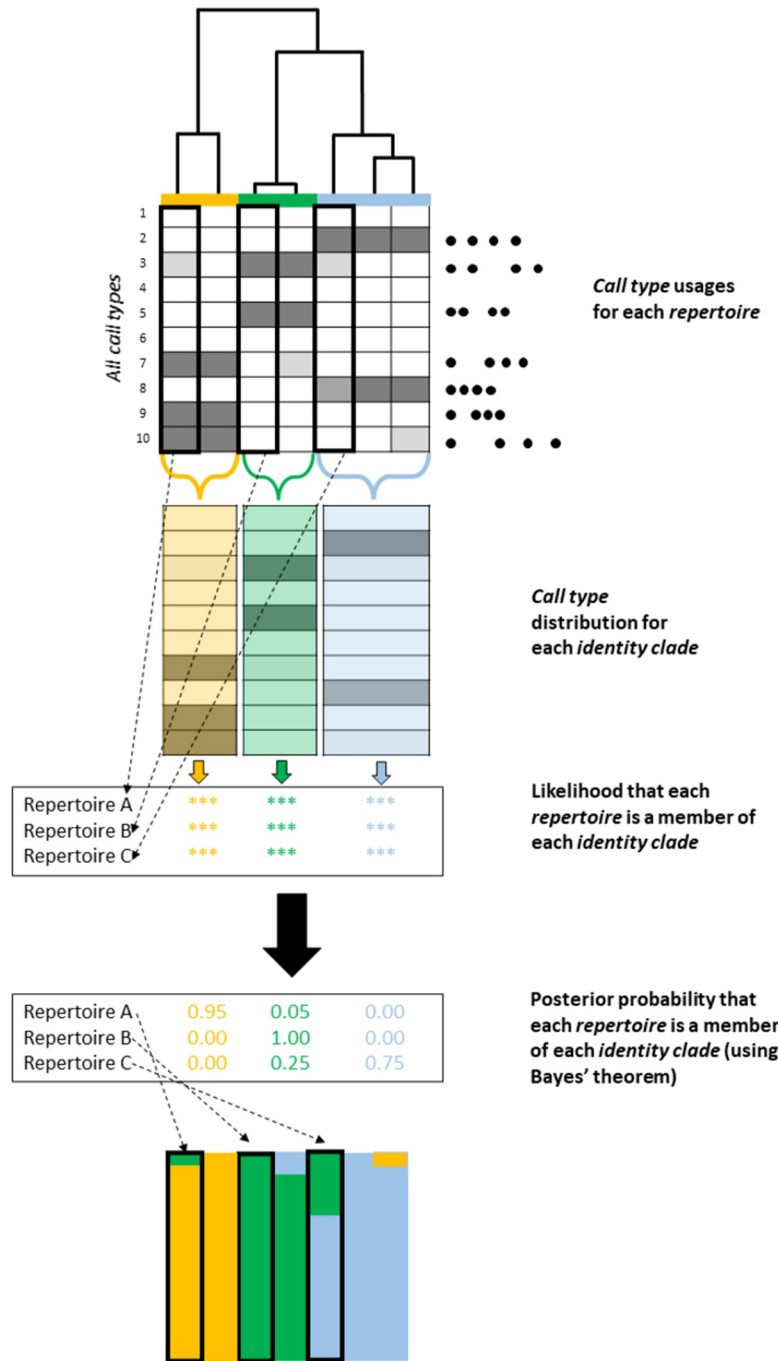

**Fig 2. The major elements of IDcallPP.** The output of IDcall is used to produce stacked barplots of the posterior probabilities that each repertoire is a member of each identity clade.

distribution of identity clade $c$ when we are addressing the likelihood that $R$ is a member of $c$, giving a revised version of Eq 2 for calls in repertoire $R$:

$$P(j, c, -R) = \frac{\sum_{r \in c; r \neq R} U(r, j)}{\sum_{r \in c; r \neq R} \sum_k U(r, k)} \tag{3}$$

Then, using the multinomial distribution, the likelihood of the distribution of call types in a repertoire $R$ given that the repertoire is a member of identity clade $c$ is:

$$L(\{U(R,j)\}|R \in c) \propto \prod_j P(j, c, -R)^{U(R,j) \cdot n(R)} \qquad (4)$$

Bayes' theorem gives the posterior probability that repertoire $R$ is a member of identity clade $c$ as:

$$Pr(R \in c) = \frac{L(\{U(R,j)\}|R \in c) \cdot Pr(R \in c)}{\sum_c L(\{U(R,j)\}|R \in c) \cdot Pr(R \in c)} \qquad (5)$$

where $Pr(R \in c)$ is the prior probability that repertoire $R$ is a member of identity clade $c$. In IDcallPP, these posterior probabilities are displayed as stacked barplots for each repertoire, as well as being output in a.csv spreadsheet file.

## Priors

There are two simple formulations for prior probabilities:

A. Equal prior probabilities of each identity clade. This is analogous to the "no admixture model" of STRUCTURE [13].

B. Prior probabilities for each identity clade are the proportion of repertoires assigned to the identity clade. This might make sense if sampling was sufficiently random or uniform (over space, time, or other relevant axes) so that the number of assignments to each identity clade was roughly proportional to its incidence in the population being considered. However, if sampling or assignation might be biased, then this option is likely inappropriate.

Other types of priors might be sensible. For instance, in the admixture model of STRUCTURE, the priors for membership of population segments are estimated using Bayesian techniques from the data itself [13]. Such formulations have yet to be implemented for IDcallPP but are a promising avenue for future development.

## Options

IdcallPP has the following options:

**Priors.** The prior probabilities of identity clade membership are either A (equal) or B (proportion of assigned repertoires), as described above. The default is A.

**Call types used.** Eq 4 can use all call types or just those found to be identity calls. In our explorations with real data (see below), we found that the "all call types" option produced clearer posterior probability plots, presumably because, in our example data sets, the non-identity calls were distributed differently among identity clades, and so provided useful information when assigning identity clade membership. This need not necessarily be the case for all data sets. However, using all call types is the default.

**Repertoire order.** The order in which the repertoires are displayed in the stacked bar plot is, by default, the order in the dendrogram plus heat map plot output from IDcall, so that the two plots can be displayed directly above one another with the repertoires lining up (see Figs 3–6). Alternatively, the input order of repertoires may be used. This could be useful if the distribution of identity clades across some axis of interest (such space or time) is desired.

Colors of identity clades: By default, the stacked bar plot of posterior probabilities uses the same color for each identity clade as in the dendrogram plus heat map plot output from IDcall (see Figs 3–7). However, these can be changed.

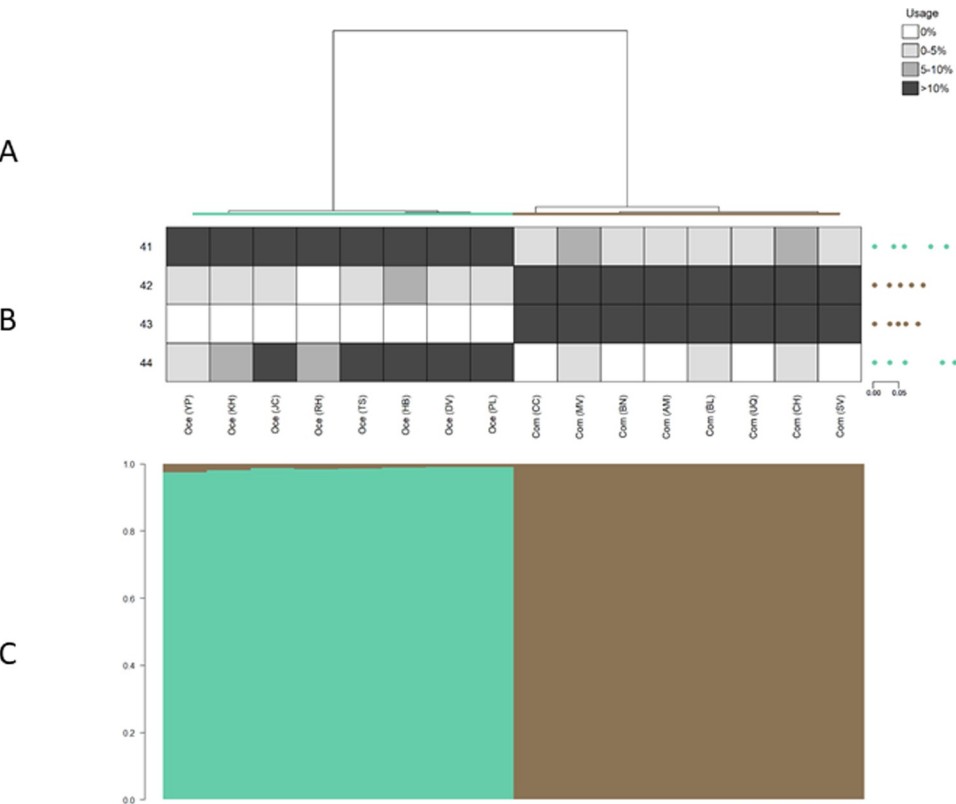

**Fig 3. Identity clades, identity song types, and posterior probabilities of repertoire assignment for crickets.**
Output from IDcall (top; taken from [14]) depicts similarity among male *Teleogryllus* cricket calling songs recorded
from individuals derived from 16 field sites in Australia (data from [17]). 'Oce' indicates song repertoires recorded
from crickets belonging to the *oceanicus* species (in teal). 'Com' denotes song repertoires recorded from crickets
belonging to the *commodus* species (in brown). The letters in parentheses denote field sites (see [17] for site
abbreviations). For each song, we created an interval vector comprised of four traits: chirp pulse length, chirp
interpulse interval, chirp-trill interval, and trill pulse length (see [17] for details on how song traits were measured).
Each repertoire (i.e. branch in the dendrogram) contains all the songs recorded from first-generation crickets that were
derived from wild-caught individuals from each field site. (A) The average linkage hierarchical clustering dendrogram
thus depicts similarity among song interval vectors of male crickets from the 16 sites. (B) The heatmap shows identity
song type usage (rows) for each field site (columns) in shades of grey, with usage calculated based on probabilistic
assignment of songs to types. Identity song type codes are on the left of the heat map and centroid song interval vector
plots are on the right (with the spaces between the dots representing chirp pulse length, chirp interpulse interval, chirp-
trill interval, and trill pulse length, and the scale bar in seconds). (C) The output from IDcallPP shows the posterior
assignment probabilities of each repertoire belonging to each identity clade (i.e. species) as a stacked bar plot. See [14]
and [17] for additional details.

## Application examples

We use the same four example acoustic data sets from three taxa as in [14]: Australian field
crickets (*Teleogryllus spp.*; hereafter crickets), grey-breasted wood-wrens (*Henicorhina leu-
cophrys*; hereafter wrens) and sperm whales (*Physeter macrocephalus*; Atlantic/Mediterranean
and Pacific datasets). These examples investigate population structure within species (sperm
whales), among subspecies (wrens), and among species (crickets). For details of call variables,
repertoire definitions, etc., see [14]. In each of Figs 3–6, we show the dendrogram plus heat
map output from IDcall [14] above the stacked bar plot of posterior probabilities of identity
clade membership from IDcallPP (using the default options listed above).

In all four example data sets, the posterior assignment probability plots from IDcallPP gen-
erally support the identity clade assignations of IDcall. The posterior probabilities for the

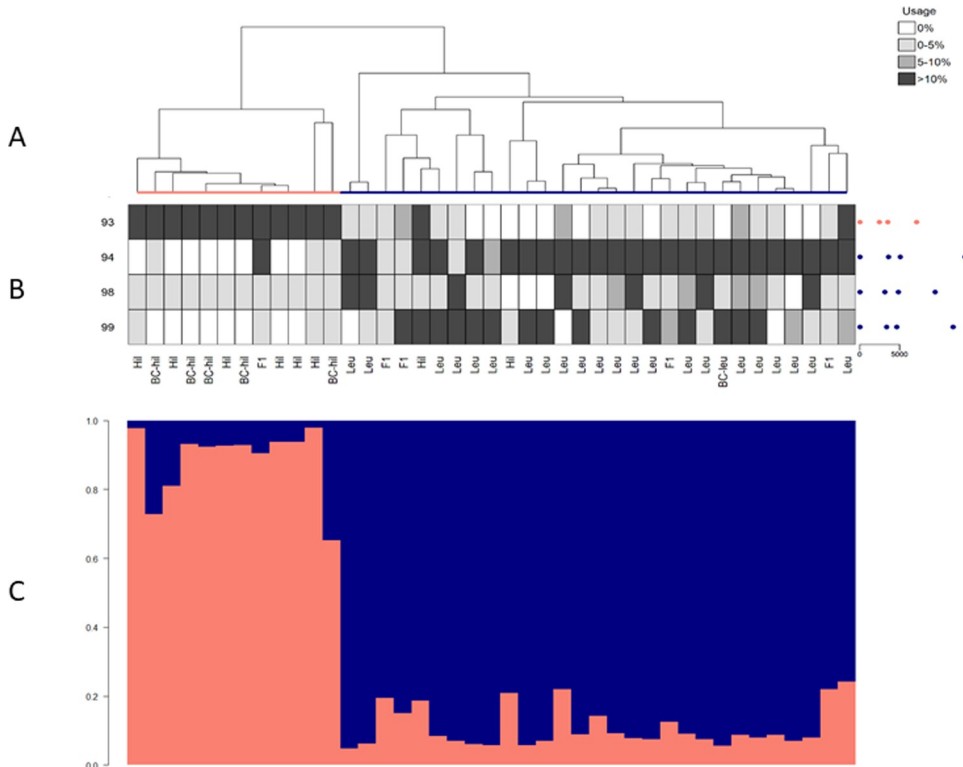

**Fig 4. Identity clades, identity song types, and posterior probabilities of repertoire assignment for wrens.** Output from IDcall (top; taken from [14]) depicts similarity among male songs (data from [18]) from two subspecies of grey-breasted wood-wren: *Henicorhina leucophrys hilaris* (salmon) and *Henicorhina leucophrys leucophrys* (navy). Genotyping abbreviations are: Hil, parental *H. l. hilaris*; Leu, parental *H. l. leucophrys*; F1, first-generation hybrid; BC-hil, backcross between Hil and F1; and BC-leu, backcross between Leu and F1. For each song, we created an interval vector comprised of three traits: averaged note peak frequency, minimum song frequency, and maximum song frequency (see [18] for details on how song traits were measured). Each repertoire (i.e. branch in the dendrogram) contains all the songs recorded from a single individual. (A) The average linkage hierarchical clustering dendrogram thus depicts similarity among song interval vectors of 41 male wrens. (B) The heatmap shows identity song type usage (rows) for each wren (columns) in shades of grey, with usage calculated based on probabilistic assignment of songs to types. Identity song type codes are on the left of the heat map and centroid song interval vector plots are on the right (with the spaces between the dots representing averaged note peak frequency, minimum song frequency, and maximum song frequency, and the scale bar in Hertz). (C) The output from IDcallPP shows the posterior assignment probabilities of each repertoire belonging to each identity clade (i.e. subspecies) as a stacked bar plot. See [14] and [18] for additional details.

cricket data set (Fig 3) shows almost perfect assignation to identity clades. It is also very good for the wren data set (Fig 4) with almost all posterior probabilities to the assigned clade greater than 0.7. The Atlantic/Mediterranean sperm whale data (Fig 5) is also very "clean" with only two repertoires having posterior probabilities to an identity clade of less than 0.7. One is a repertoire (leftmost arrow in Fig 5) that was not assigned to an identity clade; the other (rightmost arrow in Fig 5) was a repertoire that appeared on initial annotation to be a mixture of the codas from two previously described sperm whale clans (identity clades), Eastern Caribbean 1 and Eastern Caribbean 2 [15]. The posterior probabilities for the Pacific sperm whale repertoires are somewhat less clear (Fig 6). The great majority of the repertoires assigned to four of the identity clades (putative new, Four-Plus, Plus-One, and Regular clans) had posterior probabilities of >0.7 for their assigned clans. A few of the exceptions echo previous analyses. For instance, the recordings of a repertoire without a clearly dominant posterior probability (arrow in Fig 6) were from a day when photoidentification evidence indicated that there might

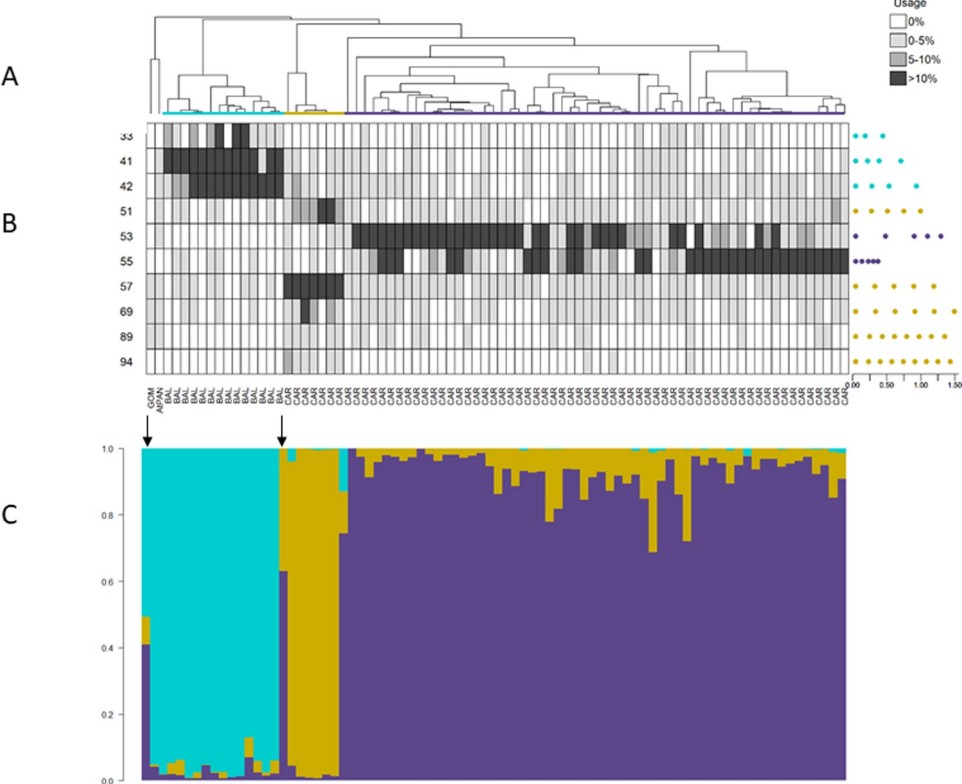

**Fig 5. Identity clades, identity coda types, and posterior probabilities of repertoire assignment for Atlantic/ Mediterranean sperm whales.** Output from IDcall (top; taken from [14]) depicts similarity among coda repertoires of sperm whale groups (data from [14]) recorded in the Atlantic Ocean and Mediterranean Sea. Colored identity clades correspond to three sperm whale clans: Mediterranean (cyan), EC2 (gold), and EC1 (purple). Location abbreviations are: AtPAN = Atlantic coast of Panama, BAL = Balearic Islands, CAR = eastern Caribbean islands, and GOM = Gulf of Mexico. Each coda was represented as a vector of inter-click intervals. Each repertoire (i.e. branch in the dendrogram) contains all the codas recorded from a known social unit of whales in a year or, if the identity of the recorded whales was unknown, all the codas recorded on a single day. (A) The average linkage hierarchical clustering dendrogram thus depicts similarity among 82 sperm whale coda repertoires. (B) The heatmap shows identity coda type usage (rows) for each repertoire (columns) in shades of grey, with usage calculated based on probabilistic assignment of codas to types. Identity coda type codes are on the left of the heat map and centroid coda interval vector plots are on the right (with the spaces between the dots representing the inter-click intervals and the scale bar in seconds). (C) The output from IDcallPP shows the posterior assignment probabilities of each repertoire belonging to each identity clade (i.e. vocal clan) as a stacked bar plot. See [14] for additional details.

be two clans present. Additionally, the repertoires assigned to the Short clan generally have much lower posterior support, which agrees with conclusions from the original IDcall analysis that the nature and structure of this identity clade were much less certain [14].

For all four of these data sets, the posterior probabilities output using the "all call types" option was clearer than when just identity calls were used (Fig 7). This indicates that while the identity calls are the primary delineators of population structure in these data sets, the other, non-identity, call types also differ somewhat in their usage among population segments.

## Discussion

IDcallPP estimates posterior probabilities that each repertoire is a member of each identity clade and provides a range of useful information. It can suggest that the population structure predicted by IDcall is extremely robust (e.g. Fig 3), robust (e.g. Fig 4), robust with an occasional, potentially interesting, outlier (e.g. Fig 5), or that parts of the population structure are

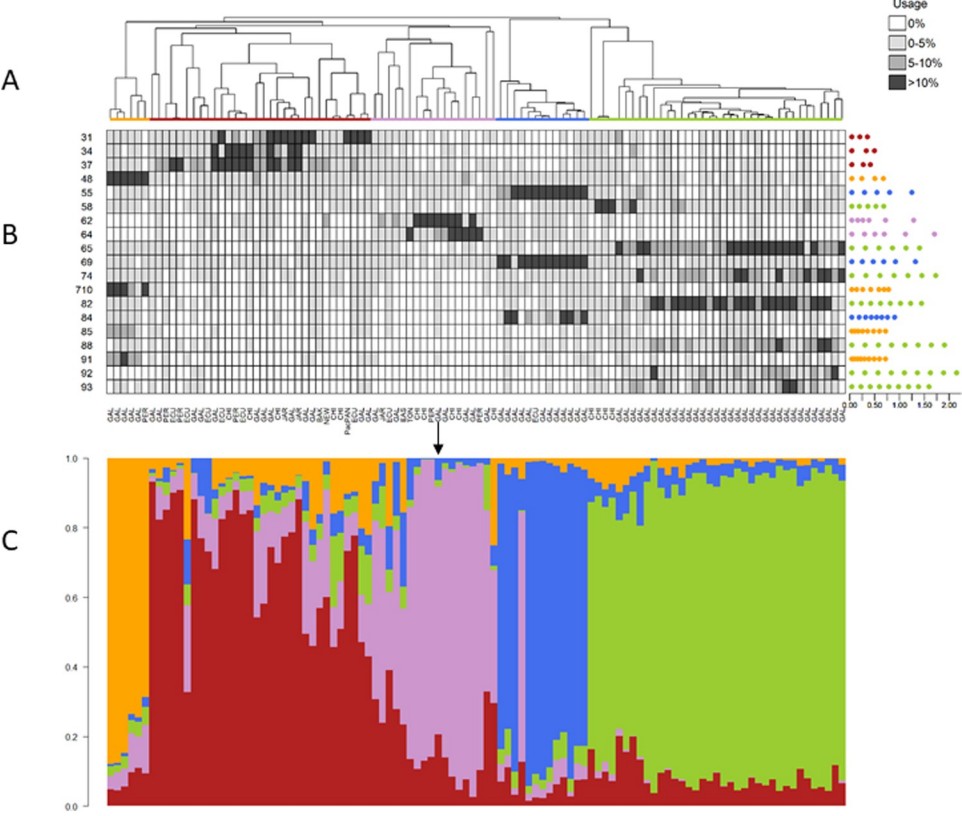

**Fig 6. Identity clades, identity coda types, and posterior probabilities of repertoire assignment for Pacific sperm whales.** Output from IDcall (top; taken from [14]) depicts similarity among coda repertoires of sperm whale groups (data from [14]) recorded in the Pacific Ocean. Colored identity clades correspond to a putative new sperm whale clan (orange) and four known clans: Short (red), Four-Plus (pink), Plus-One (blue), and Regular (green). Location abbreviations are: BAK = Baker Island, CHI = Chile, EAS = Easter Island, ECU = Ecuador, GAL = Galápagos Islands, JAR = Jarvis Island, NEW = New Zealand, PacPAN = Pacific coast of Panama, PER = Peru, and TON = Tonga. Each coda was represented as a vector of inter-click intervals. Each repertoire (i.e. branch in the dendrogram) contains all the codas recorded from a single photo-identified group of sperm whales in a year. (A) The average linkage hierarchical clustering dendrogram thus depicts similarity among 106 sperm whale coda repertoires. (B) The heatmap shows identity coda type usage (rows) for each repertoire (columns) in shades of grey, with usage calculated based on probabilistic assignment of codas to types. Identity coda type codes are on the left of the heat map and centroid coda interval vector plots are on the right (with the spaces between the dots representing the inter-click intervals and the scale bar in seconds). (C) The output from IDcallPP shows the posterior assignment probabilities of each repertoire belonging to each identity clade (i.e. vocal clan) as a stacked bar plot. See [14] for additional details.

well described while others remain unclear (e.g. Fig 6). In cases where different parameter settings for IDcall produce different population structures, it may help guide the choice of parameters.

The output stacked barplot of posterior probabilities should provide good guidance for evolutionary biologists, resource managers and conservation biologists as to the structure of their target population, in a similar way to that provided by STRUCTURE [e.g. 16]. However, the output may also address other questions of a species' biology. For instance, the relative clarity of the posterior probability plots using identity calls versus those using all calls (e.g. Fig 7) might suggest whether the acoustic signatures of identity clades are restricted to identity calls, or manifest more broadly through repertoires.

There are important differences between IDcall+IDcallPP and STRUCTURE, in addition to the different data sources (acoustic vs. genetic). Although IDcallPP calculates posterior probabilities of identity clade membership using Bayes' theorem (Eq 5), the delineation of the

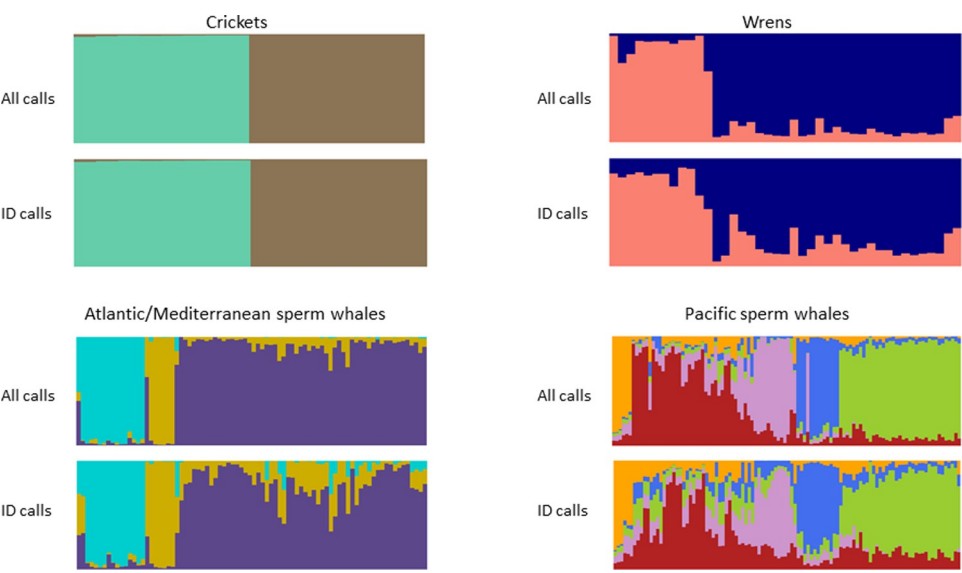

**Fig 7. Stacked barplots showing posterior probability distributions of repertoires to identity clades using all calls (above) and just identity calls (below) for the four example data sets.**

identity clades by IDcall uses a non-Bayesian, and generally more conservative, method for determining the number of population segments, and allows some repertoires not to be assigned to identity clades. It should, thus, be less prone to overestimation of the number of population segments and the misassignment of repertoires.

An issue which may affect the posterior probabilities is possible non-independence among the calls of a repertoire, thus theoretically invalidating Eq 4. We investigated the resulting biases by calculating how posterior probabilities were affected when the $\{n(R)\}$ in Eq 4 were divided by a variance inflation factor $v$, where $v>1$ indicates lack of independence in count data [17]. With two identity clades, $v = 1.2$, and a true posterior probability of 0.8 for membership of one of the identity clades this was inflated to 0.84, and with five identity clades this became 0.87. When the variance inflation factor was raised to $v = 2.0$ (indicating substantial non-independence) these posterior probabilities were raised to 0.94 and 0.98, a considerable bias upwards from 0.8. Thus, non-independence of calls may be an important issue for some data sets. A correction could be applied in situations where the variance inflation factor can be estimated.

IDcallPP employs only the no-admixture model in which a repertoire must be from only one identity clade or no identity clade at all, so the y-axes in Figs 3–6 are the posterior assignment probabilities. In the current implementation, there is no theoretical possibility that a repertoire contains elements of two or more identity clades: the posterior probabilities are that a repertoire is from a particular identity clade. However, as suggested above for the sperm whale populations, a repertoire could sometimes include calls from two, or possibly more, population segments. Thus, a useful future development would be an admixture model option in IDcallPP.

We have developed this procedure of obtaining posterior membership of population segments using the output of IDcall which delineates clades using identity calls, made often by one population segment and rarely by the others. However, posterior probabilities can be calculated whenever an acoustic data set is divided into repertoires, the elements of each repertoire can be separated into calls in a manner so that each call can be categorized or at least quantified, and then some technique is used to cluster the repertoires into population

segments. The trickiest part of this will often be calculating the likelihoods that each repertoire is a member of each population segment. If the calls can be categorized, or at least assigned probabilities of belonging to different categories (as in IDcall), and can be considered independent, then this is accomplished using Eqs 1–4. When calls are only defined by continuous measures (and not allocated to categories), one would need to obtain probability distributions for each population segment in multivariate space, perhaps using mixture models, and then assess the overlap of the calls of each repertoire with the probability distributions of each population segment.

Some of these steps could be simple. For instance calls could be allocated to call types subjectively by humans [e.g. 4] or using a simple clustering method such as K-means [18]. Population segments could be delineated geographically, or by weighting equally all the calls in each repertoire (not emphasizing identity calls as in IDcall).

Compared with molecular genetic methods for detecting, assigning, and evaluating population structure, techniques using acoustic data are much more rudimentary. They have mostly been ad-hoc methods developed or appropriated for a particular data set [e.g. 3, 18]. However, although IDcallPP has been developed to work with the output of IDcall, we have outlined a generic methodology that should be generally useful in studies of population structure using acoustic data.

The collection and analysis of acoustic data to study population structure will often be less costly, and usually less invasive, than comparable genetic studies. Sometimes, as with our wren and cricket examples, the genetic and acoustic data can tell similar stories. In contrast, when acoustic repertoires are socially learned, as with sperm whales, the contrasting patterns of genetic and cultural inheritance may lead to complex population structures [19]. Thus, the analysis of acoustic data may be effective and/or essential if we are to understand population structures.

The IDcall and IDcallPP codes (in program language R) are under active development by the authors and can be accessed, along with the sperm whale datasets, through the Open Science Framework (https://osf.io/5fter/).

## Acknowledgments

We thank Jack Rayner for directing us to the cricket dataset; Wouter Halfwerk for answering questions about the wren dataset; and Laela Sayigh, Gianni Pavan, Sara Rose, Mary Ann Daher and Kurt Fristrup for helping with the Watkins Marine Mammal Sound Database.

## Author Contributions

**Conceptualization:** Hal Whitehead, Taylor A. Hersh.

**Formal analysis:** Hal Whitehead.

**Investigation:** Hal Whitehead.

**Methodology:** Hal Whitehead, Taylor A. Hersh.

**Project administration:** Hal Whitehead.

**Resources:** Taylor A. Hersh.

**Software:** Hal Whitehead, Taylor A. Hersh.

**Supervision:** Hal Whitehead.

**Validation:** Hal Whitehead, Taylor A. Hersh.

**Writing – original draft:** Hal Whitehead.

**Writing – review & editing:** Hal Whitehead, Taylor A. Hersh.

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
