## [Decision Letter · Decision Letter 0]

21 Mar 2022

PONE-D-21-36851Posterior probabilities of membership of repertoires in acoustic cladesPLOS ONE

Dear Dr. Whitehead,

Thank you for submitting your manuscript to PLOS ONE. After careful consideration, we feel that it has merit but does not fully meet PLOS ONE’s publication criteria as it currently stands. Therefore, we invite you to submit a revised version of the manuscript that addresses the points raised during the review process.

The reviewers indicated that this is an interesting and important manuscript and I completely agree. The reviewers have a few minor comments that should be addressed by the authors. I am hoping these revisions can be made without much difficulty.

We look forward to receiving your revised manuscript.

Kind regards,

Christopher Nice, Ph.D.

Academic Editor

PLOS ONE

Journal Requirements:

Reviewers' comments:

Reviewer's Responses to Questions

**Comments to the Author**

1. Is the manuscript technically sound, and do the data support the conclusions?

Reviewer #1: Partly

Reviewer #2: Yes

2. Has the statistical analysis been performed appropriately and rigorously? 

Reviewer #1: No

Reviewer #2: Yes

3. Have the authors made all data underlying the findings in their manuscript fully available?

Reviewer #1: Yes

Reviewer #2: Yes

4. Is the manuscript presented in an intelligible fashion and written in standard English?

Reviewer #1: Yes

Reviewer #2: Yes

5. Review Comments to the Author

Reviewer #1: This is a cool paper that makes a “structure-like” model in order to identify groups that use the same call sets, in the way that Structure identifies sets of individuals with the same gene frequencies.

(1) label the axes. Most structure plots are based on admixture model, so the x axis is estimated ancestry fraction. Here this is a no admixture model, so the axis corresponds to assignment probability.

(2) several of the steps seem unclear. How is the posterior achieved? Structure uses MCMC but what is used here? It says that the model is non Bayesian. What is it instead? How is it decided which repertoires are not assigned to identity groups? It would be very instructive to compare to a fully structure like model (ie bayesian no admixure model) to justify these choices.

(3) the statement this is somewhat circular is confusing. Presumably the logic is similar to the structure paper, ie where gene frequencies can be applied given population memberships and vice versa. If this is what is meant spell it out.

(4) the approach would presumably be more powerful if IDcall and IDcallPP were combined together. It should be easier to identify different call signals given identityclade assignments. This possibility should at least be mentioned.

Reviewer #2: The paper represents a fascinating development in analyzing population structure for species that use acoustic communication. It was well written and the findings clearly presented. I did find the italicizing of keywords to be distracting after a while so I recommend not doing that. One important point that was not emphasized in the Discussion is that their method will generally be much less invasive, and possibly less costly, than genetic methods to accomplish the same task. That said a comparison to a genetic study on the same populations would be informative validation of the approach.

6. PLOS authors have the option to publish the peer review history of their article (what does this mean?). If published, this will include your full peer review and any attached files.

Reviewer #1: No

Reviewer #2: No

---

## [Author Response · Author response to Decision Letter 0]

6 Apr 2022

Thank you for considering our manuscript “Posterior probabilities of membership of repertoires in acoustic clades” for publication in PLOS ONE. We are happy with the reviews and have revised the manuscript accordingly, as follows:

Reviewer #1: This is a cool paper that makes a “structure-like” model in order to identify groups that use the same call sets, in the way that Structure identifies sets of individuals with the same gene frequencies.

(1) label the axes. Most structure plots are based on admixture model, so the x axis is estimated ancestry fraction. Here this is a no admixture model, so the axis corresponds to assignment probability.

This is a good point. Labelling the y-axes would add to the complexity of the already complex plots, so we have added (in italics) to each of the captions for Figs 3-6: “The output from IDcallPP shows the posterior assignment probabilities of each repertoire…”, and also added (lines 216-7): “…, so the y-axes in Figs 3-6 are the posterior assignment probabilities." 

(2) several of the steps seem unclear. How is the posterior achieved? Structure uses MCMC but what is used here? It says that the model is non Bayesian. What is it instead? How is it decided which repertoires are not assigned to identity groups? It would be very instructive to compare to a fully structure like model (ie bayesian no admixure model) to justify these choices. 

There is some confusion here. IDcall (the original Hersh et al. methodology) does not use Bayesian methods to find identity calls or identity clades, but the methodology used in the submitted paper (IDcallPP) to produce posterior probabilities does (equation 5 is Bayes’ theorem). In this case the calculation is straightforward, so methods like MCMC are not needed. To make this clearer, we have added (lines 205-6) “Although IDcallPP calculates posterior probabilities of identity clade membership using Bayes’ theorem (eq 5), the delineation of the identity clades by IDcall uses a non-Bayesian, and generally more conservative, method…”

(3) the statement this is somewhat circular is confusing. Presumably the logic is similar to the structure paper, ie where gene frequencies can be applied given population memberships and vice versa. If this is what is meant spell it out.

We have spelled this out, adding (lines 73-75): “In other words, the calls heard in a repertoire are used to delineate identity clades, the very information that is used to calculate the posterior probability that the repertoire is a member of an identity clade. To remove this circularity, we omit repertoire R from the calculation…”

(4) the approach would presumably be more powerful if IDcall and IDcallPP were combined together. It should be easier to identify different call signals given identity clade assignments. This possibility should at least be mentioned.

Yes, they are now combined. We have added (lines 242-3): “The IDcall and IDcallPP codes (in program language R) are under active development by the authors and can be accessed, along with the sperm whale datasets, through the Open Science Framework (https://osf.io/5fter/).” In the latest version of software, IDcall optionally calls IDcallPP.

Reviewer #2: The paper represents a fascinating development in analyzing population structure for species that use acoustic communication. It was well written and the findings clearly presented. I did find the italicizing of keywords to be distracting after a while so I recommend not doing that. 

We have removed the italicizing of key words as suggested by the reviewer.

One important point that was not emphasized in the Discussion is that their method will generally be much less invasive, and possibly less costly, than genetic methods to accomplish the same task. That said a comparison to a genetic study on the same populations would be informative validation of the approach.

We have added (lines 237-241): “The collection and analysis of acoustic data to study population structure will often be less costly, and usually less invasive, than comparable genetic studies. Sometimes, as with our wren and cricket examples, the genetic and acoustic data can tell similar stories. In contrast, when acoustic repertoires are socially learned, as with sperm whales, the contrasting patterns of genetic and cultural inheritance may lead to complex population structures [18]. Thus, the analysis of acoustic data may be effective and/or essential if we are to understand population structures.”

---

## [Editor Report · Decision Letter 1]

11 Apr 2022

Posterior probabilities of membership of repertoires in acoustic clades

PONE-D-21-36851R1

Dear Dr. Whitehead,

We’re pleased to inform you that your manuscript has been judged scientifically suitable for publication and will be formally accepted for publication once it meets all outstanding technical requirements.

Kind regards,

Christopher Nice, Ph.D.

Academic Editor

PLOS ONE
---

## [Editor Report · Acceptance letter]

14 Apr 2022

PONE-D-21-36851R1 

Posterior probabilities of membership of repertoires in acoustic clades 

Dear Dr. Whitehead:

I'm pleased to inform you that your manuscript has been deemed suitable for publication in PLOS ONE. Congratulations! Your manuscript is now with our production department. 

Kind regards, 

on behalf of

Dr. Christopher Nice 

Academic Editor

PLOS ONE